# HOGWILD!-Gibbs Can Be PanAccurate

**Constantinos Daskalakis** [*]
EECS & CSAIL, MIT
costis@csail.mit.edu

**Nishanth Dikkala** [*]
EECS & CSAIL, MIT
nishanthd@csail.mit.edu

**Siddhartha Jayanti** [*†]
EECS & CSAIL, MIT
jayanti@mit.edu

## Abstract

Asynchronous Gibbs sampling has been recently shown to be fast-mixing and an accurate method for estimating probabilities of events on a small number of variables of a graphical model satisfying Dobrushin's condition [DSOR16]. We investigate whether it can be used to accurately estimate expectations of functions of *all the variables* of the model. Under the same condition, we show that the synchronous (sequential) and asynchronous Gibbs samplers can be coupled so that the expected Hamming distance between their (multivariate) samples remains bounded by $O(\tau \log n)$, where $n$ is the number of variables in the graphical model, and $\tau$ is a measure of the asynchronicity. A similar bound holds for any constant power of the Hamming distance. Hence, the expectation of any function that is Lipschitz with respect to a power of the Hamming distance, can be estimated with a bias that grows logarithmically in $n$. Going beyond Lipschitz functions, we consider the bias arising from asynchronicity in estimating the expectation of polynomial functions of all variables in the model. Using recent concentration of measure results [DDK17, GLP17, GSS18], we show that the bias introduced by the asynchronicity is of smaller order than the standard deviation of the function value already present in the true model. We perform experiments on a multi-processor machine to empirically illustrate our theoretical findings.

## 1 Introduction

The increasingly ambitious applications of data analysis, and the corresponding growth in the size of the data that needs to processed has brought important scalability challenges to machine learning algorithms. Fundamental methods such as Gradient Descent and Gibbs sampling, which were designed with a sequential computational model in mind, are to be applied on datasets of increasingly larger size. As such, there has recently been increased interest towards developing techniques for parallelizing these methods. However, these algorithms are inherently sequential and are difficult to parallelize.

HOGWILD!-SGD, proposed by Niu et al. [NRRW11], is a lock-free asynchronous execution of stochastic gradient descent that has been shown to converge under the right sparsity conditions. Several variants of this method, and extensions of the asynchronous execution approach have been recently proposed, and have found successful applications in a broad range of applications ranging from PageRank approximation, to deep learning and recommender systems [YHSD12, NO14, MBDC15, MPP+15, LWR+15, LWR+15, DSZOR15].

Similar to HOGWILD!-SGD, lock-free asynchronous execution of Gibbs sampling, called HOGWILD!-Gibbs, was proposed by Smola and Narayanamurthy [SN10], and empirically shown to work well on several models [ZR14]. Johnson et al. [JSW13] provide sufficient conditions under

---

[*]Supported by NSF awards CCF-1617730 and IIS-1741137, a Simons Investigator Award, a Google Faculty Research Award, and an MIT-IBM Watson AI Lab research grant.

[†]Also supported by the Department of Defense (DoD) through the NDSEG Program.

which they show theoretically that HOGWILD!-Gibbs produces samples with the correct mean in Gaussian models, while Terenin et al. [TSD15] propose a modification to the algorithm that is shown to converge under some strong assumptions on asynchronous computation.

**Input:** Set of variables $V$, Configuration $x_0 \in S^{|V|}$, Distribution $\pi$
initialization;
**for** $t = 1$ **to** $T$ **do**
    Sample $i$ uniformly from $\{1, 2, \ldots, n\}$;
    Sample $X_i \sim \Pr_\pi [.|X_{-i} = x_{-i}]$ and set $x_{i,t} = X_i$;
    For all $j \neq i$, set $x_{j,t} = x_{j,t-1}$;
**end**

**Algorithm 1:** Gibbs Sampling

In a more recent paper, De Sa et al. [DSOR16] propose the study of HOGWILD!-Gibbs under a stochastic model of asynchronicity in graphical models with discrete variables. Whenever the graphical model satisfies Dobrushin's condition, they show that the mixing time of the asynchronous Gibbs sampler is similar to that of the sequential (synchronous) one. Moreover, they establish that the asynchronous Gibbs sampler accurately estimates probabilities of events on a sublinear number of variables, in particular events on up to $O(\varepsilon n / \log n)$ variables can be estimated within variational distance $\varepsilon$, where $n$ is the total number of variables in the graphical model (Lemma 2, [DSOR16]).

**Our Results.**   Our goal in this paper is to push the theoretical understanding of HOGWILD!-Gibbs to estimate functions of *all the variables in a graphical model*. In particular, we are interested in whether HOGWILD!-Gibbs can be used to accurately estimate the expectations of such functions. Results from [DSOR16] imply that an accurate estimation is possible whenever the function under consideration is Lipschitz with a good Lipschitz constant with respect to the Hamming metric. Under the same Dobrushin condition used in [DSOR16] (see Definition 3), and under a stochastic model of asynchronicity with weaker assumptions (see Section 2.1), we show that you can do better than the bounds implied by [DSOR16] even for functions with bad Lipschitz constants. For instance, consider quadratic functions on an Ising model, which is a binary graphical model, and serves as a canonical example of Markov random fields [LPW09, MS10, Fel04, DMR11, GG86, Ell93]. Under appropriate normalization, these functions take values in the range $[-n^2, n^2]$ and have a Lipschitz constant of $n$. Given this, the results of [DSOR16] would imply we can estimate quadratic functions on the Ising model within an error of $O(n)$. We improve this error to be of $O(\sqrt{n})$. In particular, we show the following in our paper:

- Starting at the same initial configuration, the executions of the sequential and the asynchronous Gibbs samplers can be coupled so that the expected Hamming distance between the multivariate samples that the two samplers maintain is bounded by $O(\tau \log n)$, where $n$ is the number of variables in the graphical model, and $\tau$ is a measure of the average contention in the asynchronicity model of Section 2.1. See Lemma 2. More generally, the expectation of the $d$-th power of the Hamming distance is bounded by $C(d, \tau) \log^d n$, for some function $C(d, \tau)$. See Lemma 3.

- It follows from Lemmas 2 and 3 that, if a function $f$ of the variables of a graphical model is $K$-Lipschitz with respect to the $d$-th power of the Hamming distance, then the bias in the expectation of $f$ introduced by HOGWILD!-Gibbs under the asynchronicity model of Section 2.1 is bounded by $K \cdot C(d, \tau) \log^d n$. See Corollary 1.

- Next, we improve the bounds of Corollary 1 for functions that are degree-$d$ polynomials of the variables of the graphical model. Low degree polynomials on graphical models are a natural class of functions which are of interest in many statistical tasks performed on graphical models (see, for instance, [DDK18] ). For simplicity we show these improvements for the Ising model, but our results are extendible to general graphical models. We show, in Theorem 4, that the bias introduced by HOGWILD!-Gibbs in the expectation of a degree-$d$ polynomial of the Ising model is bounded by $O((n \log n)^{(d-1)/2})$. This bound improves upon the bound computed by Corollary 1 by a factor of about $(n/ \log n)^{(d-1)/2}$, as the Lipschitz constant with respect to the Hamming distance of a degree-$d$ polynomial of the Ising model can be up to $O(n^{d-1})$. Importantly, the bias of $O((n \log n)^{(d-1)/2})$ that we show is introduced by the asynchronicity is of a lower order of magnitude than the standard de-

viation of degree-$d$ polynomials of the Ising model, which is $O((n)^{d/2})$—see Theorem 2, and which is already experienced by the sequential sampler. Moreover, in Theorem 5, we also show that the asynchronous Gibbs sampler is not adding a higher order variance to its sample. Thus, our results suggest that running Gibbs sampling asynchronously leads to a valid bias-variance tradeoff.

Our bounds for the expected Hamming distance between the sequential and the asynchronous Gibbs samplers follow from coupling arguments, while our improvements for polynomial functions of Ising models follow from a combination of our Hamming bounds and recent concentration of measure results for polynomial functions of the Ising model [DDK17, GLP17, GSS18].

- In Section 5, we illustrate our theoretical findings by performing experiments on a multi-core machine. We experiment with graphical models over two kinds of graphs. The first is the $\sqrt{n} \times \sqrt{n}$ grid graph (which we represent as a torus for degree regularity) where each node has 4 neighbors, and the second is the clique over $n$ nodes.

  We first study how valid the assumptions of the asynchronicity model are. The main assumption in the model was that the average contention parameter $\tau$ doesn't grow as the number of nodes in the graph grows. It is a constant which depends on the hardware being used and we observe that this is indeed the case in practice. The expected contention grows linearly with the number of processors on the machine but remains constant with respect to $n$ (see Figures 1 and 2).

  Next, we look at quadratic polynomials over graphical models associated with both the grid and clique graphs. We estimate their expected values under the sequential Gibbs sampler and HOGWILD!-Gibbs and measure the bias (absolute difference) between the two. Our theory predicts that this should scale at $\sqrt{n}$ and we observe that this is indeed the case (Figure 3). Our experiments are described in greater detail in Section 5.

## 2 The Model and Preliminaries

In this paper, we consider the Gibbs sampling algorithm as applied to discrete graphical models. The models will be defined on a graph $G = (V, E)$ with $|V| = n$ nodes and will represent a probability distribution $\pi$. We use $S$ to denote the range of values each node in $V$ can take. For any configuration $X \in S^{|V|}$, $\pi_i(.|X^{-i})$ will denote the conditional distribution of variable $i$ given all other variables of state $X$.

In Section 4, we will look at Ising models, a particular class of discrete binary graphical models with pairwise local correlations. We consider the Ising model on a graph $G = (V, E)$ with $n$ nodes. This is a distribution over $\Omega = \{\pm 1\}^n$, with a parameter vector $\vec{\theta} \in \mathbb{R}^{|V|+|E|}$. $\vec{\theta}$ has a parameter corresponding to each edge $e \in E$ and each node $v \in V$. The probability mass function assigned to a string $x$ is

$$P(x) = \exp\left(\sum_{v \in V} \theta_v x_v + \sum_{e=(u,v) \in E} \theta_e x_u x_v - \Phi(\vec{\theta})\right),$$

where $\Phi(\vec{\theta})$ is the log-partition function for the distribution. We say an Ising model has *no external field* if $\theta_v = 0$ for all $v \in V$. For ease of exposition we will focus on the case with no external field in this paper. However, the results extend to Ising models with external fields when the functions under consideration (in Section 4) are appropriately chosen to be *centered*. See [DDK17].

Throughout the paper we will focus on bounded functions defined on the discrete space $S^{|V|}$. For a function $f$, we use $\|f\|_\infty$ to denote the maximum absolute value of the function over its domain. We will use $[n]$ to denote the set $\{1, 2, \ldots, n\}$. In Section 4, we will study polynomial functions over the Ising model. Since $x_i^2 = 1$ always in an Ising model, any polynomial function of degree $d$ can be represented as a multilinear function of degree $d$ and we will refer to them interchangeably in the context of Ising models.

**Definition 1** (Polynomial/Multilinear Functions of the Ising Model). *A degree-$d$ polynomial defined on $n$ variables $x_1, \ldots, x_n$ is a function of the following form*

$$\sum_{S \subseteq [n]:|S| \leq d} a_S \prod_{i \in S} x_i,$$

*where $a : 2^{[n]} \rightarrow \mathbb{R}$ is a coefficient vector.*

We will use $a$ to denote the coefficient vector of such a multilinear function and $\|a\|_\infty$ to denote the maximum element of $a$ in absolute value. Note that we will use permutations of the subscripts to refer to the same coefficient, i.e., $a_{ijk}$ is the same as $a_{jik}$.

We now give a formal definition of Dobrushin's uniqueness condition, also known as the high-temperature regime. First we define the influence of a node $j$ on a node $i$.

**Definition 2** (Influence in Graphical Models)**.** *Let $\pi$ be a probability distribution over some set of variables $V$. Let $B_j$ denote the set of state pairs $(X, Y)$ which differ only in their value at variable $j$. Then the influence of node $j$ on node $i$ is defined as*

$$I(j, i) = \max_{(X,Y) \in B_j} \left\| \pi_i(. | X^{-i}) - \pi_i(. | Y^{-i}) \right\|_{TV}$$

Now, we are ready to state Dobrushin's condition.

**Definition 3** (Dobrushin's Uniqueness Condition)**.** *Consider a distribution $\pi$ defined on a set of variables $V$. Let*

$$\alpha = \max_{i \in V} \sum_{j \in V} I(j, i)$$

*$\pi$ is said to satisfy Dobrushin's uniqueness condition if $\alpha < 1$.*

We have the following result from [DSOR16] about mixing time of Gibbs sampler for a model satisfying Dobrushin's condition.

**Theorem 1** (Mixing Time of Sequential Gibbs Sampling)**.** *Assume that we run Gibbs sampling on a distribution that satisfies Dobrushin's condition, $\alpha < 1$. Then the mixing time of sequential-Gibbs is bounded by*

$$t_{mix-seq(\varepsilon)} \leq \frac{n}{1 - \alpha} \log\left(\frac{n}{\varepsilon}\right).$$

**Definition 4.** *For any discrete state space $S^{|V|}$ over the set of variables $V$, The* Hamming distance *between $x, y \in S^{|V|}$ is defined as $d_H(x, y) = \sum_{i \in V} \mathbb{1}_{\{x_i \neq y_i\}}$.*

**Definition 5** (The greedy coupling between two Gibbs Sampling chains)**.** *Consider two instances of Gibbs sampling associated with the same discrete graphical model $\pi$ over the state space $S^{|V|}$: $X_0, X_1, \ldots$ and $Y_0, Y_1, \ldots$. The following coupling procedure is known as the* greedy coupling. *Start chain 1 at $X_0$ and chain 2 at $Y_0$ and in each time step $t$, choose a node $v \in V$ uniformly at random to update in both the chains. Without loss of generality assume that $S = \{1, 2, \ldots, k\}$. Let $p(i_1)$ denote the probability that the first chain sets $X_{t,v} = i_1$ and let $q(i_2)$ be the probability that the second chain sets $Y_{t,v} = i_2$. Plot the points $\sum_{j=1}^{i} p(j) = P(i)$, and $\sum_{j=1}^{i} q(j) = Q(i)$ for all $i \in [k]$ on the interval from $[0, 1]$. Also pick $P(0) = Q(0) = 0$ and $P(k + 1) = Q(k + 1) = 1$. Couple the updates according to the following rule:*

*Draw a number $x$ uniformly at random from $[0, 1]$. Suppose $x \in [P(i_1), P(i_1 + 1)]$ and $x \in [Q(i_2), Q(i_2 + 1)]$. Choose $X_{t,v} = i_1$ and $Y_{t,v} = i_2$.*

We state an important property of this coupling which holds under Dobrushin's condition, in the following Lemma.

**Lemma 1.** *The greedy coupling (Definition 5) satisfies the following property. Let $X_0, Y_0 \in S^{|V|}$ and consider two executions of Gibbs sampling associated with distribution $\pi$ and starting at $X_0$ and $Y_0$ respectively. Suppose the executions were coupled using the greedy coupling. Suppose in the step $t = 1$, node $i$ is chosen to be updated in both the models. Then,*

$$\Pr\left[X_{1,i} \neq Y_{1,i}\right] \leq \left\| \pi_i(. | X_0^{-i}) - \pi_i(. | Y_0^{-i}) \right\|_{TV} \tag{1}$$

## 2.1 Modeling Asynchronicity

We use the asynchronicity model from [RRWN11] and [DSOR16]. Hogwild!-Gibbs is a multi-threaded algorithm where each thread performs a Gibbs update on the state of a graph which is

stored in shared memory (typically in RAM). We view each processor's write as occuring at a distinct time instant. And each write starts the next time step for the process. Assuming that the writes are all serialized, one can now talk about the state of the system after $t$ writes. This will be denoted as time $t$. HOGWILD! is modeled as a stochastic system adapted to a natural filtration $\mathcal{F}_t$. $\mathcal{F}_t$ contains all events thast have occured until time $t$. Some of these writes happen based on a read done a few steps ago and hence correspond to updates based on stale values in the local cache of the processor. The staleness is modeled in a stochastic manner using the random variable $\tau_{i,t}$ to denote the delay associated with the read performed on node $i$ at time step $t$. The value of node $i$ used in the update at time $t$ is going to be $Y_{i,t} = X_{i,(t-\tau_{i,t})}$. Delays across different node reads can be correlated. However delay distribution is independent of the configuration of the model at time $t$. The model imposes two restrictions on the delay distributions. First, the expected value of each delay distribution is bounded by $\tau$. We will think of $\tau$ as a constant compared to $n$ in this paper. We call $\tau$ the average contention parameter associated with a HOGWILD!-Gibbs execution. [DSOR16] impose a second restriction which bounds the tails of the distribution of $\tau_{i,t}$. We do not need to make this assumption in this paper for our results. [DSOR16] need the assumption to show that the HOGWILD! chain mixes fast. However, by using coupling arguments we can avoid the need to have the HOGWILD! chain mix and will just use the mixing time bounds for the sequential Gibbs sampling chain instead. Let T denote the set of all delay distributions. We refer to the sequential Gibbs sampler associated with a distribution $\pi$ as $G_\pi$ and the HOGWILD! Gibbs sampler together with T associated with a distribution $p$ by $H_p^{\mathrm{T}}$. Note that $H_\pi$ is a time-inhomogenuous Markov chain and might not converge to a stationary distribution.

## 2.2 Concentration of Polynomials on Ising Models

Here we state a known result about concentration of measure for polynomial functions on Ising models satisfying Dobrushin's condition.

**Theorem 2** (Concentration of Measure for Polynomial Functions of the Ising model, [DDK17, GLP17, GSS18]). *Consider an Ising model $p$ without external field on a graph $G = (V, E)$ satisfying Dobrushin's condition with Dobrushin parameter $\alpha < 1$. Let $f_a$ be a degree $d$-polynomial over the Ising model. Let $X \sim p$. Then, there is a constant $c(\alpha, \delta)$, such that,*

$$\Pr\left[|f_a(X) - \mathbf{E}\left[f_a(X)\right]| > t\right] \leq 2\exp\left(-\frac{(1-\alpha)t^{2/d}}{c(\alpha, d)\|a\|_\infty^{2/d}n}\right).$$

*As a corollary this also implies,*

$$\mathbf{Var}\left[f_a(X)\right] \leq C_3(d, \alpha)n^d.$$

## 3 Bounding The Expected Hamming Distance Between Coupled Execution of HOGWILD! and Sequential Gibbs Samplers

In this Section, we show that under the greedy coupling of the sequential and asynchronous chains, the expected Hamming distance between the two chains at any time $t$ is small. This will form the basis for our accurate estimation results of Section 4. We begin by with Lemma 2.

**Lemma 2.** *Let $\pi$ denote a discrete probability distribution on $n$ variables (nodes) with Dobrushin parameter $\alpha < 1$. Let $G_\pi = X_0, X_1, \ldots, X_t, \ldots$ denote the execution of the sequential Gibbs sampler on $\pi$ and $H_\pi^{\mathrm{T}} = Y_0, Y_1, \ldots, Y_t, \ldots$ denote the HOGWILD! Gibbs sampler associated with $\pi$ such that $X_0 = Y_0$. Suppose the two chains are running coupled in a greedy manner. Let $\mathcal{K}_t$ denote all events that have occured until time $t$ in this coupled execution. Then we have, for all $t \geq 0$, under the greedy coupling of the two chains,*

$$\mathbf{E}\left[d_H(X_t, Y_t)|\mathcal{K}_0\right] \leq \frac{\tau\alpha\log n}{1-\alpha}$$

At a high level, the proof proceeds by studying the expected change in the Hamming distance under one step of the coupled execution of the chains. We can bound the expected change using the Dobrushin parameter and the property of the greedy coupling (Lemma 1). We then show that the expected change is negative whenever the Hamming distance between the two chains was above

$O(\log n)$ to begin with. This allows us to argue that when the two chains start at the same configuration, then the expected Hamming distance remains bounded by $O(\log n)$.

Next, we generalize the above Lemma to bound also the $d^{th}$ moment of the Hamming distance between $X_t$ and $Y_t$ obtained from the coupled executions.

**Lemma 3** ($d^{th}$ moment bound on Hamming). *Consider the same setting as that of Lemma 2. We have, for all $t \geq 0$, under the greedy coupling of the two chains,*

$$\mathbf{E}\left[d_H(X_t, Y_t)^d | \mathcal{K}_0\right] \leq C(\tau, \alpha, d) \log^d n,$$

*where $C(.)$ is some function of the parameters $\tau, \alpha$ and $d$.*

The proof of Lemma 3 follows a similar flavor as that of Lemma 2. It is however more involved to bound the expected increase in the $d^{th}$ power of the Hamming distance and it requires some careful analysis to see that the bound doesn't scale polynomially in $n$.

## 4 Estimating Global Functions Using HOGWILD! Gibbs Sampling

To begin with, we observe that our Hamming moment bounds from Section 3 imply that we can accurately estimate functions or events of the graphical model if they are Lipschitz. We show this below as a Corollary of Lemma 3.

Now, we state Corollary 1 which quantifies the error we can attain when trying to estimate expectations of Lipschitz functions using HOGWILD!-Gibbs.

**Corollary 1.** *Let $\pi$ denote the distribution associated with a graphical model over the set of variables $V$ ($|V| = n$) taking values in a discrete space $S^n$. Assume that the model satisfies Dobrushin's condition with Dobrushin parameter $\alpha < 1$. Let $f : S^{|V|} \to \mathbb{R}$ be a function such that, for all $x, y \in S^{|V|}$,*

$$|f(x) - f(y)| \leq K d_H(x, y)^d.$$

*Let $X \sim \pi$ and let $Y_0, Y_1, \ldots, Y_t$ denote an execution of HOGWILD!-Gibbs sampling on $\pi$ with average contention parameter $\tau$. For $t > \frac{n}{1-\alpha} \log\left(2 \|f\|_\infty n/K\right)$,*

$$|\mathbf{E}[f(Y_t)] - \mathbf{E}[f(X)]| \leq K.(C(\tau, \alpha, d) \log^d n + 1).$$

We note that the results of [DSOR16] can be used to obtain Corollary 1 when the function is Lipschitz with respect to the Hamming distance. The above corollary provides a simple way to bound the bias introduced by HOGWILD! in estimation of Lipschitz functions. However, many functions of interest over graphical models are not Lipschitz with good Lipschitz constants. In many cases, even when the Lipschitz constants are bad, there is still hope for more accurate estimation. As it turns out Dobrushin's condition provides such cases. We will focus on one such case which is polynomial functions of the Ising model. Our goal will be to accurately estimate the expected values of constant degree polynomials over the Ising model. Using the bounds from Lemmas 2 and 3, we now proceed to bound the bias in computing polynomial functions of the Ising model using HOGWILD! Gibbs sampling.

We first remark that linear functions (degree 1 polynomials) suffer 0 bias in their expected values due to HOGWILD!-Gibbs. This is because under zero external field Ising models $\mathbf{E}[\sum_i a_i X_i] = 0$ since each node individually has equal probability of being $\pm 1$. This symmetry is maintained by HOGWILD!-Gibbs since the delays are configuration-agnostic. Hence the delays when a node is $+1$ and when it is $-1$ can be coupled perfectly leaving the symmetry intact. Therefore, we start our investigation at quadratic polynomials. Theorem 3 states the bound we show for the bias in computation of degree 2 polynomials of the Ising model.

**Theorem 3** (Bias in Quadratic functions of Ising Model computed using HOGWILD!-Gibbs). *Consider the quadratic function $f_a(x) = \sum_{i,j:i<j} a_{ij} x_i x_j$. Let $p$ denote an Ising model on $n$ nodes with Dobrushin parameter $\alpha < 1$. Let $\{X_t\}_{t \geq 0}$ denote a run of sequential Gibbs sampler and $H_p^{\mathrm{T}} = \{Y_t\}_{t \geq 0}$ denote a run of HOGWILD!- Gibbs on $p$, such that $X_0 = Y_0$. Then we have, for $t > \frac{6n}{1-\alpha} \log(2 \|a\|_\infty n)$, under the greedy coupling of the two chains,*

$$|\mathbf{E}[f_a(X_t) - f_a(Y_t)]| \leq c_2 \|a\|_\infty \frac{\tau \alpha \log n}{(1-\alpha)^{3/2}} (n \log n)^{1/2}.$$

The main intuition behind the proof is that we can improve upon the bound implied by the Lipschitz constant by appealing to strong concentration of measure results about functions of graphical models under Dobrushin's condition [DDK17, GLP17, GSS18].

We extend the ideas in the above proof to bound the bias introduced by the HOGWILD! Gibbs algorithm when computing the expected values of a degree $d$ polynomial of the Ising model in high temperature. Our main result concerning $d$-linear functions is Theorem 4.

**Theorem 4** (Bias in degree $d$ polynomials computed using HOGWILD!-Gibbs)**.** *Consider a degree $d$ polynomial of the form $f_a(x) = \sum_{i_1 i_2,\ldots,i_d} a_{i_1 i_2 \ldots i_d} x_{i_1} x_{i_2} \ldots x_{i_d}$. Consider the same setting as that of Theorem 3. Then we have, for $t > \frac{n(d+1)}{1-\alpha} \log n$, under the greedy coupling of the two chains,*

$$|\mathbf{E}[f_a(X_t) - f_a(Y_t)]| \leq c' \, \|a\|_\infty \, (n \log n)^{(d-1)/2}.$$

Next, we show that we can accurately estimate the expectations above by showing that the variance of the functions under the asynchronous model is comparable to that of the functions under the sequential model.

**Theorem 5** (Variance of degree $d$ polynomials computed using HOGWILD!-Gibbs)**.** *Consider a high temperature Ising model $p$ on $n$ nodes with Dobrushin parameter $\alpha < 1$. Let $f_a(x)$ be a degree $d$ polynomial function Let $Y_0, Y_1, \ldots, Y_t$ denote a run of HOGWILD! Gibbs sampling associated with $p$. We have, for $t > \frac{(d+1)n}{1-\alpha} \log (n^2)$,*

$$\mathbf{Var}\,[f(Y_t)] \leq \|a\|_\infty^2 \, C(d, \alpha, \tau) n^d.$$

## 4.1 Going Beyond Ising Models

We presented results for accurate estimation of polynomial functions over the Ising model. However, the results can be extended to hold for more general graphical models satisfying Dobrushin's condition. A main ingredient here was concentration of measure. If the class of functions we look at has $d^{th}$-order bounded differences in expectation, then we indeed get concentration of measure for these functions (Theorem 1.2 of [GSS18]). This combined with the techniques in our paper would allow similar gains in accurate estimation of such functions on general graphical models.

## 5 Experiments

We show the results of experiments run on a machine with four 10-core Intel Xeon E7-4850 CPUs to demonstrate the practical validity of our theory. In our experiments, we focused on two Ising models—Curie-Weiss and the Grid. The Curie-Weiss $CW(n, \alpha)$ is the Ising model corresponding to the complete graph on $n$ vertices with edges of weight $\beta = \frac{\alpha}{n-1}$. The $Grid(k^2, \alpha)$ model is the Ising model corresponding to the $k$-by-$k$ grid with the left connected to the right and top connected to the bottom to form a torus—a four-regular graph; the edge weights are $\frac{\alpha}{4}$. The total influence of each of these models is at most $\alpha$, so we chose $\alpha = 0.5$ to ensure Dobrushin's condition. To generate samples, we start at a uniformly random configuration and run Markov chains for $T = 10n \log_2(n)$ steps to ensure mixing.

In our first experiment (Figure 1) we validate the modeling assumption that the average delay of a read $\tau$ is a constant. Computing the exact delays in a real run of the HOGWILD! is not possible, but we approximate the delays by making processes log read and write operations to a lock-free queue as they execute the HOGWILD!-updates. We present two plots of the average delay of a read in a HOGWILD! run of the $CW(n, 0.5)$ Markov chain with respect to $n$. Four asynchronous processors were used to generate the first plot, while twenty were used for the second. We notice that the average delay depends on the number of asynchronous processes, but is constant with respect to $n$ as assumed in our model.

Next, we plot (in Figure 2) the relationship between the number of asynchronous processors used in a HOGWILD! execution and the delay parameter $\tau$. For this plot, we estimated $\tau$ by the average empirical delay over HOGWILD! runs of $CW(n, 0.5)$ models, with $n$ ranging from 100 to 1000 in increments of one hundred. The plot shows a linear relationship, and suggests that the delay per additional processor is approximately $0.4$ steps.

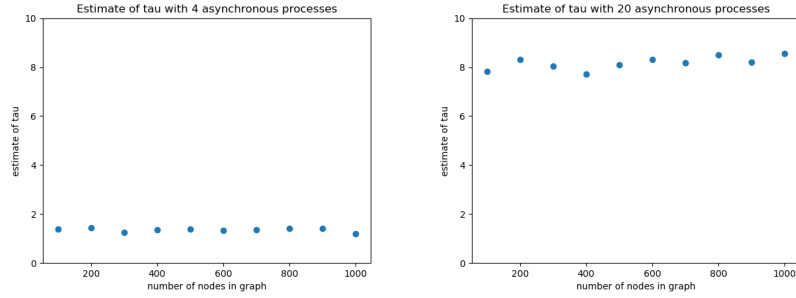

Figure 1: Average delay of reads for $CW(n, 0.5)$ model. Four asynchronous processors were used on the left, while twenty were used on the right.

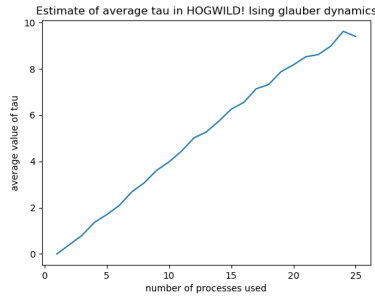

Figure 2: Average delay of reads for $CW(n, 0.5)$ model as the number of processors used varies.

The primary purpose of our work is to demonstrate that polynomial statistics computed from samples of a HOGWILD! run of Gibbs Sampling will approximate those computed from a sequential run. Our third experiment demonstrates exactly this fact. We plot (in Figure 3 on the left) the empirical expectations of the *complete bilinear function* $f(X_1, \ldots, X_n) = \sum_{i \neq j} X_i X_j$ as we vary the number of nodes $n$ in a Curie-Weiss model graph. Each red point is the empirical mean of the function $f$ computed over 5000 samples from the HOGWILD! Markov chain corresponding to $CW(n, 0.5)$, and each blue point is the empirical mean produced from 5000 sequential runs of the same chain. Our theory (Theorem 3) predicts that the bias, the vertical difference in height between red and blue points, at any given value of $n$ will be on the order of the standard deviation divided by $\sqrt{n}$ (standard deviation is $\Theta(n)$ and bias is $O(\sqrt{n})$). We plot error bars of this order, and find that the HOGWILD! means fall inside the error bars, thus corroborating our theory. We show that theory and practice coincide even for sparse graphs, by making the same plot for the $Grid(n, 0.5)$ model on the right of the same figure.

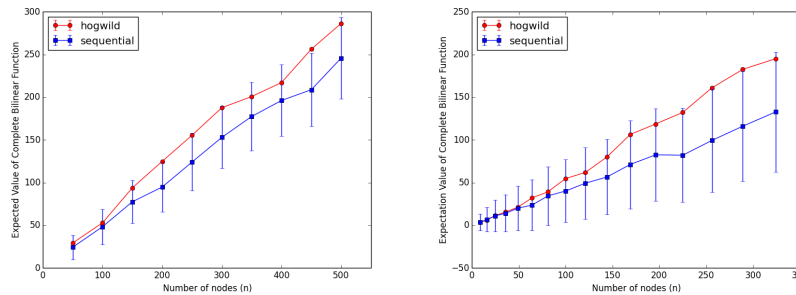

Figure 3: Means (with appropriately scaled error bars) of the complete bilinear function computed over 5000 sequential and hogwild runs of $CW(n, 0.5)$ (left) and $Grid(n, 0.5)$ (right).

# 6 Acknowledgements

We thank Prof. Srinivas Devdas and Xiangyao Yu for helping us gain access to and program on their multicore machines.

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
