[Reviews · NeurIPS 2018]

Reviewer 1



The authors prove theorems about the accuracy of asynchronous Gibbs sampling in graphical models with discrete variables that satisfy Dobrushin's condition. I am not familiar with this literature, so I'm taking the authors' description of the state of the literature as a given. The authors' results are as follows (let n be the number of variables in the graphical model, let t be the time index, and let tau be the maximum expected read delay in the asynchronous sampler): - Lemma 2. The asynchronous Gibbs sampler can be coupled to a synchronous Gibbs sampler with the same initial state such that the expected Hamming distance between them is bounded by O(tau*log(n)) uniformly in t. Lemma 3 gives an analogous bound for the dth moment of the Hamming distance. - Corollary 1. If a function f is K-Lipschitz with respect to the dth power of the Hamming distance, the bias of the asynchronous Gibbs sampler for the expectation of f is bounded by log^d(n) (plus a constant, times a constant, and for sufficiently large t). Previous results (De Sa et al., 2016) matched this bound for d=1. - Theorem 4. For an Ising model with no external field, still satisfying Dobrushin's condition, the expectation of a degree-d polynomial f under asynchronous Gibbs, say E(f(Y_t)), minus the expectation of the same polynomial under synchronous Gibbs, say E(f(X_t)), is bounded by O((n log n)^((d-1)/2)) (for sufficiently large t). Previous results (De Sa et al., 2016) implied O(n) for d=2. The supplementary material contains the full paper with proofs (which I haven't checked carefully). The main paper, with the exception of the introduction and experiments, has been imperfectly excerpted from the full paper and does not flow well in some portions (for example, lines 220-224). I'd recommend rereading the main paper to ensure that it flows well. In the experiments section, the authors run synchronous and asynchronous Gibbs to provide empirical support for Theorem 4. The authors plot the difference between the expectations of a bilinear function under synchronous and asynchronous Gibbs as a function of n; they overlay the curve sqrt(n) (predicted by Theorem 4) and proclaim it a match (see Figure 3). The dots are noisy and don't look particularly like sqrt(n) to me. Besides, the theoretical result is only an upper bound; there's no reason to think it should match. line 143: It's a result about synchronous Gibbs, but you've written HOGWILD!-Gibbs and t_{mix-hog}; this should be changed. To summarize, it seems this paper is providing somewhat improved bounds for the bias of asynchronous Gibbs under similar conditions as previous work. The presentation is reasonably clear, and the topic is timely.

Reviewer 2



Author-feedback response: Author response did clarify many of my concerns. Thus, I increased my rating. But please make sure to clarify"synchronous Gibbs sampling" (by either renaming to traditional "sequential" or properly defining). ----------------------------------------------------------------------------------------------------------- This paper attempts to analyze asynchronous Gibbs sampling for models satisfying Dobrushin’s condition. The paper builds upon previous seminal work by Chris De Sa et al. (2016). Informally, the key idea behind all the results in the paper is to try to show that sequential and asynchronous chains are similar. In particular, they begin by showing that expected Hamming distance between an asynchronous Gibbs chain and its sequential version starting from the state is small at any time. Using this they could bound the bias in estimating the expected value of polynomials of the random variables. Their bounds improved upon previous results. Finally, some empirical studies on simple test cases were provided. Comments: 1. The writing and clarity of paper can be improved. The paper should be scanned for consistency and confusing order of symbols. The main confusion arises from nomenclature used. Please correct me if I am wrong, does synchronous sampling mean sequential sampling in this paper? To me synchronous sampling means sampling all variables together and updating the state and this strategy is known to not work for many cases (one example below). I think that in the paper, this is not the case, rather it refers to sequential sampling and my review is based on this assumption. Also, in Theorem 1 (borrowed from Chris De Sa et al. 2016), on one hand, its labeled as “Mixing Time of Synchronous Gibbs Sampling” but on the other hand, theorem statement says “mixing time of HOGWILD!-Gibbs”. Please fix/clarify this. 2. With the assumptions of asynchronicity mentioned in the paper, consider a trivial 2D Ising model (this example can be generalized to the Ising model on an n-dimensional torus). I believe the following read-write sequence …, R_1, R_2, W_1, W_2, R1, R2, W2, W1 … is possible. To elaborate, in this case, all processors read the current state and then both writes are completed before next reads. But, we know the stationary distribution of this sampler in closed form and it’s different from the target Ising distribution. (c.f. Neumann, A.U., and Derrida, B., 1988. Finite-size scaling study of dynamical phase transitions in two-dimensional models: Ferromagnet, symmetric and non-symmetric spin glasses. Journal de Physique, 49(10), pp.1647-1656 and Zaheer, M., Wick, M., Tristan, J.B., Smola, A. and Steele, G., 2016, May. Exponential stochastic cellular automata for massively parallel inference. In Artificial Intelligence and Statistics (pp. 966-975).) Can you please explain what assumption of your result is broken by this example that I am missing? 3. The experiments were basic, but verifying the claim. The experiments can be made more interesting by analyzing corner cases (like the example above) or testing limits of asynchronicity etc. Overall, the paper takes a step in our understanding of asynchronous Gibbs sampling. The proof strategies are mostly standard, but improving the writing/presentation and providing extensive empirical evaluations would strengthen the paper a lot, making it a nice contribution to the field. Minor Comments/Typos: 1. Apologies for nit-picking: In the introduction section, in my humble opinion the contribution section is too long (It’s ok for a journal paper, but quite a stretch for an 8 page NIPS paper). My personal preference is to limit the contributions in the introduction to major points at a high level. The details about how the obtained results are an improvement over previous work can be part of the main text. 2. A lot of space in the experiment section can be saved. This can allow you to write more details about the proof techniques in the main paper. 3. Line 141: ... condition is alpha < 1. <=> ... condition if alpha < 1. 4. Eq(10)-(11) in appendix: Placement of \alpha is confusing wrt log.

Reviewer 3



The paper proposes an improvement over the bias bounds of asynchronous Gibbs sampling over those in the recent paper by C. De Sa et al (2016); the authors show that the Hamming distance between samples from Hogwild and synchronous Gibbs chains are O(log n) under certain coupling constructions. They also analyze the bias for polynomial function expectations on Ising models using recent results on concentration of measure. Empirical results on a couple of synthetic Ising models provide evidence in favor of tight coupling of these expectations between Hogwild and synchronous Gibbs runs. Quality ===== Overall quality of work is pretty good. The coupling construction in Definition 5 is simple and elegant, and the theoretical results are solid. The treatment of the Hamming distance bounds for the polynomial Ising functions in \S 3 is very comprehensive. I did not verify the proofs in the supplementary in detail, but the theorem statements seem intuitively correct. Clarity ===== I found the paper to be well written, albeit somewhat dense. Minor issues: - In Definition 5, given the setup on lines 156--157,perhaps it would be clearer if X_{t,v} and Y_{t,v} were set to different values in lines 151--152 to avoid confusion. - line 50: whenver -> whenever Originality ======== Moderate. The main contributions of the paper are the coupling construction for the sampling chains and the ensuing bounds on the Hamming distances and biases. While the theoretical analysis is very thorough, it is basically an improvement of the bounds in C. De Sa et al's work, and thus not very novel in my opinion. Significance ========== - The theoretical contribution of the paper is its main strength; the bounds on the polynomial functions should be useful for further work in analyzing mixing rates, as it appears they can be combined with Taylor approximations to derive bounds for more general functions. - While the coupling construction is elegant, the assumption of the a Hogwild run runs updating the same variables at every time step as a standard synchronous sampler seems very strong to me. - A more detailed experimental section would have made the paper stronger; in particular, error bounds for the Hogwild runs in Figure 2 would be informative. Overall I think the paper makes a solid contribution to mixing rate analyses of asynchronous Gibbs samplers, and will be of interest to the community.